# Cellular and Molecular Effects of Eribulin in Preclinical Models of Hematologic Neoplasms

**DOI:** 10.3390/cancers14246080

**Published:** 2022-12-10

**Authors:** Hugo Passos Vicari, Keli Lima, Leticia Veras Costa-Lotufo, João Agostinho Machado-Neto

**Affiliations:** 1Department of Pharmacology, Institute of Biomedical Sciences, University of São Paulo, São Paulos 05508-000, Brazil; 2Laboratory of Medical Investigation in Pathogenesis and Targeted Therapy in Onco-Immuno-Hematology (LIM-31), Department of Internal Medicine, Hematology Division, Faculdade de Medicina, University of São Paulo, São Paulo 01246-903, Brazil

**Keywords:** eribulin, microtubule dynamics, acute leukemias, drug resistance

## Abstract

**Simple Summary:**

Hematologic neoplasms comprise a heterogeneous group of diseases that interfere with normal blood production. Treating patients who fail available therapies for these diseases is an ongoing challenge. Thus, the search for new treatment options is urgent, and drug repositioning is emerging as an attractive strategy for finding new effective drugs. Eribulin is a drug that acts on microtubules, and it is used in solid tumors, and its safety is known. In the present study, we provide evidence of the effects of eribulin on hematologic cancers and identify the potential biomarkers of responsiveness. Our study indicates that eribulin is a candidate blood cancer drug for repositioning.

**Abstract:**

Despite the advances in understanding the biology of hematologic neoplasms which has resulted in the approval of new drugs, the therapeutic options are still scarce for relapsed/refractory patients. Eribulin is a unique microtubule inhibitor that is currently being used in the therapy for metastatic breast cancer and soft tissue tumors. Here, we uncover eribulin’s cellular and molecular effects in a molecularly heterogeneous panel of hematologic neoplasms. Eribulin reduced cell viability and clonogenicity and promoted apoptosis and cell cycle arrest. The minimal effects of eribulin observed in the normal leukocytes suggested selectivity for malignant blood cells. In the molecular scenario, eribulin induces DNA damage and apoptosis markers. The *ABCB1*, *ABCC1*, p-AKT, p-NFκB, and NFκB levels were associated with responsiveness to eribulin in blood cancer cells, and a resistance eribulin-related target score was constructed. Combining eribulin with elacridar (a P-glycoprotein inhibitor), but not with PDTC (an NFkB inhibitor), increases eribulin-induced apoptosis in leukemia cells. In conclusion, our data indicate that eribulin leads to mitotic catastrophe and cell death in blood cancer cells. The expression and activation of MDR1, PI3K/AKT, and the NFκB-related targets may be biomarkers of the eribulin response, and the combined treatment of eribulin and elacridar may overcome drug resistance in these diseases.

## 1. Introduction

Hematologic neoplasms comprise a heterogeneous group of diseases that interfere with the normal production of blood cells, among which, acute leukemias stand out for their high mortality and recurrence rates [1]. Despite the great advances in understanding the biology of these diseases which has resulted in the approval of new drugs, the therapeutic options are still scarce for the relapsed/refractory patients [2,3].

Microtubules are essential components for cell division due to their contractile properties, and they are considered one of the most important molecular targets for cancer treatment. Microtubule-targeting agents suppress microtubule dynamics and present variation in terms of the modulation (stabilizing and destabilizing activity) and binding sites to tubulin [4,5,6]. Indeed, antimicrotubule therapy has been widely used, and it has significantly contributed to cancer therapy over the past 50 years. The vinca alkaloids (vincristine, vinblastine, vindesine, vinorelbine, and vinflunine) represent the oldest class of microtubule-interfering agents, and they have been used to treat a wide range of malignancies, including leukemias, lymphomas, and solid tumors [7]. Paclitaxel and its semi-synthetic analog docetaxel have been utilized for treating advanced ovarian, breast, lung, head and neck, and prostate cancers [8].

In the context of hematologic malignancies, vinca alkaloids are extensively used in the treatment of acute lymphoblastic leukemia (ALL) [9], with vincristine being one of the few drugs that is approved for the treatment of relapsing T-ALL [10]. In addition, several tubulin inhibitors are in ongoing clinical trials against acute myeloid leukemia (AML) [11].

Despite the considerable number of microtubule-modulating agents available, the mechanisms that limit the effectiveness of the current treatments, such as the overexpression of specific tubulin isoforms, mutations, and other mechanisms of resistance which have not yet been elucidated, are a significant clinical problem, and this reinforces the need for the search for new and more selective agents that may act against resistant cells [12,13,14].

Recently, a unique microtubule inhibitor, eribulin, has drawn attention due to its anticancer properties. It is currently being used in metastatic breast cancer and soft tissue tumor therapy [15,16], but its effects on blood cancers have been underexplored. Here, we show that eribulin provides potent antineoplastic effects in a molecularly heterogeneous panel of blood cancer cells, and we uncover the underlying anticancer molecular mechanisms.

## 2. Materials and Methods

### 2.1. Healthy Donors’ Samples

Peripheral blood mononuclear cells (PBMCs) were obtained from eight healthy donors by Ficoll-Hypaque density gradient centrifugation (Sigma-Aldrich, St. Louis, MO, USA) according to the manufacturer’s instructions. All of the procedures were approved by the Ethics Committee of the Institute of Biomedical Sciences of the University of São Paulo (CAAE: 39510920.1.0000.5467). Informed consent was obtained from all of the donors, and all of the methods were conducted in accordance with the Declaration of Helsinki. The PBMCs were cultured in RPMI-1640 medium containing 30% fetal bovine serum (FBS), penicillin/streptomycin, and recombinant cytokines (30 ng/mL IL3, 100 ng/mL IL7, 100 ng/mL FLT3-ligand, and 30 ng/mL SCF, PeproTech, Rocky Hill, NJ, USA) at a density of 2 × 10^6^ cells/mL.

### 2.2. Cell Culture and Chemical Reagents

The OCI-AML3, Kasumi-1, HL-60, THP-1, MOLM-13, MV4-11, NB4, and NB4-R2 cells were kindly provided by Prof. Eduardo Magalhães Rego (University of São Paulo, Ribeirão Preto, Brazil). The U-937, K-562, KU812, HEL, Jurkat, Namalwa, Daudi, Raji, U266, MM1.S, and MM1.R cells were kindly provided by Prof. Sara Teresinha Olalla Saad (University of Campinas, Campinas, Brazil). The CEM, NALM6, and REH were kindly provided by Dr. Gilberto Carlos Franchi Junior (University of Campinas, Campinas, Brazil). The SET-2 cells were kindly provided by Prof. Fabíola Attié de Castro (University of São Paulo, Ribeirão Preto, Brazil). Karpas 442 was kindly provided by Prof. Martin Dreyling (University Hospital Grosshadern/LMU, Munich, Germany). SUP-B15 was kindly provided by Dr. Lucas Eduardo Botelho de Souza (National Institute of Science and Technology in Stem Cells and Cell Therapy, Ribeirão Preto, São Paulo, Brazil).

The cells were cultured in an appropriate media (RPMI-1640, IMDM, or alpha-MEM) supplemented with 10 or 20% FBS according to the American Type Culture Collection (ATCC) or Deutsche Sammlung von Mikroorganismen und Zellkulturen (DSMZ) recommendations, plus 1% penicillin/streptomycin. The cell cultures were maintained at 5% CO_2_ and 37 °C. Eribulin mesylate was purchased from Sigma-Aldrich.

### 2.3. Cell Viability Assay

A total of 2 × 10^4^ cells (cell lines) or 2 × 10^5^ (PBMC) per well were seeded in 96-well plates in the appropriate medium in the presence of a vehicle or different concentrations of eribulin (0, 0.032, 0.16, 0.8, 4, 20, or 100 nM for 72 h or 0, 0.062, 0.125, 0.25, 0.5, 1, or 2 nM for 24, 48, and 72 h). Next, 10 μL methylthiazoletetrazolium (MTT) solution (5 mg/mL) was added and incubated at 37 °C in 5% CO_2_ for 4 h. The reaction was stopped with 150 μL 0.1 N HCl in anhydrous isopropanol. The cell viability was evaluated by measuring the absorbance at 570 nm. The IC_50_ values were calculated using nonlinear regression analysis in GraphPad Prism 5 (GraphPad Software, Inc., San Diego, CA, USA).

### 2.4. Apoptosis Assay

A total of 1 × 10^5^ cells per well were seeded in 24-well plates in the presence of a vehicle or eribulin (0.25, 0.5, and 1 nM) for 72 h. The cells were then washed twice with ice-cold PBS and resuspended in a binding buffer containing 1 μg/mL propidium iodide (PI), and 1 μg/mL APC-labeled annexin V (BD Biosciences, San Jose, CA, USA). All of the specimens were acquired by flow cytometry (FACSCalibur; Becton Dickinson, Franklin Lakes, NJ, USA) after incubation for 15 min at room temperature in a light-protected area, and they were analyzed using FlowJo software (Treestar, Inc., San Carlos, CA, USA).

### 2.5. Cell Cycle Analysis

A total of 4 × 10^5^ cells per well were seeded in six-well plates in the presence of a vehicle or eribulin (0.25, 0.5, and 1 nM), harvested at 72 h, fixed with 70% ethanol, and stored at 4 °C for at least 2 h before the analysis. The fixed cells were stained with 20 μg/mL PI containing 10 μg/mL RNase A for 30 min at room temperature in a light-protected area. The DNA content distribution was acquired in a FACSCalibur cytometer (Becton Dickinson), and it was analyzed using FlowJo software (Treestar, Inc.).

### 2.6. Cellular Morphology Analysis

The leukemia cells were treated with a vehicle or eribulin (0.25, 0.5, or 1 nM) for 72 h. For the morphology analysis, the vehicle- or eribulin-treated cells (1 × 10^5^) were adhered to microscopic slides using cytospin (Serocito, Model 2400, FANEM, Guarulhos, Brazil) for the subsequent Rosenfeld staining. The morphological analyses of the nucleus and cytoplasm of the treated cells were visualized using a Leica DM 2500 optical microscope, and the images were acquired using the LAS V4.6 software (Leica, Benshein, Germany).

### 2.7. Colony Formation Assay

Colony formation was conducted in a semi-solid methylcellulose medium (0.5 × 10^3^ cells/mL; MethoCult 4230; StemCell Technologies Inc., Vancouver, BC, Canada). The NB4, NB4-R2, MOLM3, OCI-AML3, Jurkat, and Namalwa cells were seeded in the presence of a vehicle or eribulin (0.12, 0.25, 0.5, or 1 nM) for eight days. The colonies were detected by adding 100 µL (5 mg/mL) of MTT reagent, and they were scored using the Image J quantification software (U.S. National Institutes of Health, Bethesda, MD, USA).

### 2.8. Western Blot Analysis

The cells were treated with a vehicle or eribulin (0.25, 0.5, and 1 nM) for 72 h, and the total protein was extracted using a buffer containing 100 mM Tris (pH 7.6), 1% Triton X-100, 2 mM PMSF, 10 mM Na_3_VO_4_, 100 mM NaF, 10 mM Na_4_P_2_O_7_, and 4 mM EDTA. Equal amounts of protein (30 μg) were then subjected to SDS-PAGE, which was followed by electrotransfer to nitrocellulose membrane. The membranes were blocked with 5% milk, incubated with specific primary antibodies diluted in blocking buffer, and then, they were incubated with secondary antibodies conjugated to HRP (horseradish peroxidase). A Western blot analysis with the indicated primary antibodies was performed using the SuperSignal West Dura Extended Duration Substrate System (Thermo Fisher Scientific, San Jose, CA, USA) and a G: BOX Chemi XX6 gel document system (Syngene, Cambridge, UK). The antibodies against stathmin 1 (OP18; sc-55531), p-stathmin 1^S16^ (p-OP18 Ser16; sc-12948-R), and γH2AX (sc-517348) were obtained from Santa Cruz Biotechnology (Santa Cruz, CA, USA). The antibodies against PARP1 (#9542) and α-tubulin (#2144) were from Cell Signaling Technology (Danvers, MA, USA). The band intensities were measured using the UN-SCAN-IT gel 6.1 software (Silk Scientific; Orem, UT, USA). The data were illustrated using multiple experiment viewer (MeV) 4.9.0 software [17]. The cropped gels retained the important bands, and the whole gel images are available in Appendix A.

### 2.9. Quantitative PCR (qPCR)

The total RNA was obtained using TRIzol reagent (Thermo Fisher Scientific). The cDNA was synthesized from 1 µg of RNA using a High-Capacity cDNA Reverse Transcription Kit (Thermo Fisher Scientific). Quantitative PCR (qPCR) was performed using a QuantStudio 3 Real-Time PCR System in conjunction with a SybrGreen System and specific primers (Appendix A). *HPRT1* and *ACTB* were used as the reference genes. The relative quantification value was calculated using Equation 2^−ΔΔCT^ [18]. A negative ‘No Template Control’ was included for each primer pair. The dissociation protocol was performed at the end of each run to check for non-specific amplification. The data were illustrated using MeV 4.9.0 software [17].

### 2.10. Statistical Analysis

The statistical analysis was performed using GraphPad Instat 5 (GraphPad Software, Inc.). An ANOVA test and a Bonferroni post-test were used for comparisons, and the Spearman test was used for the correlation analysis. All of the *p* values were two-sided with a significance level of 5%.

## 3. Results

### 3.1. Eribulin Reduces Cell Viability of Hematologic Neoplasm-Derived Cell Lines

Firstly, the antineoplastic activity of eribulin was evaluated in a large panel of myeloid and lymphoid cancer cell lines. Eribulin displays cytotoxic activity in the nanomolar range (IC_50_ values ranged from 0.13 to 12.12 nM), and only five out of twenty-one blood cancer cell lines were considered to be resistant to the drug (IC_50_ > 100 nM). Among the myeloid models, the cell lines with constitutive tyrosine kinase activation caused by the mutations BCR::ABL1 and JAK2^V617F^ (K-562, KU812, SET-2, and HEL) presented higher IC_50_ values for eribulin (from 12.12 to >100 nM). Among the lymphoid models, the cell lines derived from multiple myeloma (U266, MM1.S, and MM1.R) presented higher IC_50_ values for eribulin (from 10.66 to 37.03 nM) (Figure 1A and Appendix A). In the normal leukocytes, eribulin did not impact the cell viability, suggesting selectivity for malignant blood cells and a favorable therapeutic window (Figure 1A). Based on these findings, six acute leukemia cells were selected for a further analysis (NB4, NB4-R2, OCI-AML3, MOLM-13, Jurkat, and Namalwa). As shown in Figure 1B, eribulin exhibited dose- and time-dependent cytotoxicity in all of the leukemia cells assessed.

### 3.2. Eribulin Promotes Apoptosis and Cell Cycle Arrest and Reduces Clonogenicity in Acute Leukemia Cells

Next, the cellular mechanisms triggered by eribulin in the leukemia cells were investigated. Eribulin significantly induced concentration-dependent apoptosis in the leukemia cells, as observed by phosphatidylserine exposure (all *p* < 0.05, Figure 2), and it increased the subG_1_ cell populations (all *p* < 0.05, Figure 3). Eribulin also induced cell cycle arrest at the G_2_/M phase for the NB4, NB-R2, OCI-AML3, and Jurkat cells and at the G_0_/G_1_ phase for MOLM-13 (all *p* < 0.05, Figure 3). The morphological analyses revealed a high frequency of mitotic aberrations upon conducting the treatment with eribulin for 72 h (Figure 4), which corroborates the cell cycle-related findings. Similarly, the long-term exposure to eribulin markedly decreased the autonomous clonal growth in the NB4, NB4-R2, MOLM-13, OCI-AML3, and Jurkat cells (all *p* < 0.05, Figure 5). The Namalwa cells were less sensitive to eribulin, which is consistent with the initial cell viability assays.

### 3.3. Eribulin Induces Molecular Markers of DNA Damage and Apoptosis in Acute Myeloid and Lymphoid Leukemia Cells

At the molecular level, the analyses of STMN1 and its inactive form (p-STMN1^S16^) (proliferation marker), γH2AX (DNA damage marker), and the cleaved PARP1 (apoptosis marker) levels were evaluated. At low eribulin concentrations, an induction of STMN1 phosphorylation was observed, while at high concentrations, the STMN1 expression was downregulated (all *p* < 0.05). Additionally, the eribulin exposure significantly induced γH2AX expression and PARP1 cleavage in all of the cell lines analyzed (Figure 6).

### 3.4. P-Glycoprotein and NFκB-Mediated Pathways Are Related to Eribulin Resistance in Hematologic Malignancies

Next, we evaluated whether the already known response biomarkers for solid tumors could be applied in the context of hematologic malignancies [19,20,21,22,23,24]. First, we determined the expression of the genes encoding the transmembrane drug efflux pump (ABCB1 and ABCC1), which encodes tubulin beta 3 class III (TUBB3) and STMN1 (Figure 7A) and the markers PI3K/AKT and NFkB-mediated signaling pathway expression and activation (Figure 7B,C). Among these molecular targets, ABCB1 (r = 0.53, *p* = 0.02), ABCC1 (r = 0.67, *p* = 0.001), p-AKT (r = 0.53, *p* = 0.02), p-NFκB (r = 0.80, *p* < 0.0001), and NFκB (r = 0.81, *p* < 0.0001) were associated with IC_50_ values for eribulin in the blood cancer cell lines (Figure 7D). Based on these data, we constructed a resistance eribulin-related target (RERT) score that precisely predicted the sensitivity of the drug in the cellular models evaluated (Figure 7E).

Finally, using the pharmacological inhibitors of P-glycoprotein (elacridar) and NFκB (PDTC), we observed that the resistance conferred by the drug efflux pump, but not NFκB activation, is reversible, and it can eliminate a leukemia cell model with the activation of both of the molecular processes (Figure 8). Combining eribulin and elacridar was not toxic to the normal leukocytes (Appendix A).

## 4. Discussion

Here, we investigated eribulin’s cellular and molecular effects in a molecularly heterogeneous panel of hematologic neoplasm cell lines, including acute myeloid leukemia, myeloproliferative neoplasms, acute lymphoblastic leukemia, multiple myeloma, and lymphoma models. Eribulin is a simplified synthetic analog of halichondrin B, a molecule isolated from a rare marine sponge *Halichondria okadai* [25,26]. It is currently approved for pre-treated and anthracycline- and taxane-refractory patients with metastatic breast cancer and metastatic liposarcoma [27,28]. In the context of hematologic malignancies, our study is pioneering, and it opens the possibility of using eribulin for the patients with these diseases.

Eribulin is a microtubule inhibitor, and it exerts its anticancer property primarily through the inhibition of tubulin and mitosis [29,30,31,32]. Unlike other antimitotic drugs, such as vinblastine and paclitaxel, which attenuate the shortening and growth phases of dynamic microtubule instability, eribulin inhibits microtubule growth by a final poisoning mechanism [29]. Thus, it does not cause the shortening of the tubulin, but it transforms them into non-productive aggregates [29].

In the present study, eribulin presented a high cytotoxic effect in blood cancer cells with minimal impact on the normal leukocytes. In agreement, eribulin has been reported to be a powerful chemotherapeutic agent with a low-to-moderate toxicity profile [33]. The cellular phenotype observed for the eribulin treatment in the leukemia cells indicated that the drug’s primary cellular mechanism of action is conserved: cell cycle blockage in G_2_/M and cell death after prolonged and irreversible mitotic arrest [29,34,35,36].

Molecularly, eribulin reduced the expression and activity of STMN1, and induced PARP1 cleavage and H2AX phosphorylation. STMN1 functions as a marker of normal and malignant hematopoietic cell proliferation, and it plays a key role in cell cycle progression and clonogenicity in acute leukemia cells [37,38,39]. Recently, STMN1 expression has been indicated as a molecular target of eribulin and associated with the response to the drug [24]. PARP1 is one of the cellular substrates of caspases, and when it is cleaved and inactivated by active caspases 3 and 7, it is considered to be a hallmark of apoptosis, forming 24 kDa and 89 kDa fragments [40,41]. H2AX is a component of the histone octamer in nucleosomes that, in the presence of DNA damage, are phosphorylated at serine 139 (i.e., γH2AX), thus making it a DNA damage marker [42]. Overall, the molecular scenario of treating acute leukemia cells with eribulin yielded reduced cell proliferation, apoptosis, and DNA damage.

Despite their importance to the antineoplastic arsenal, antimicrotubule agents present various resistance mechanisms that may lead to treatment failure and reduced survival rates [43]. Thus, understanding eribulin-related resistance mechanisms may improve the response rates. Over the last few years, several mechanisms have been reported, particularly the activation of P-glycoprotein and pathways mediated by PI3K/AKT and NFκB [19,20,21,22,23,44]. Indeed, in the present study, the sensitivity to eribulin was associated with the *ABCB1*, *ABCC1*, p-AKT, p-NFκB, and NFκB expression levels in the blood cancer cells.

The multidrug resistance (MDR) phenotype is a challenge in the therapeutic management of several neoplasms since cancer cells become unresponsive to many anticancer drugs [45,46]. Several cellular and molecular mechanisms mediate this complex process, and one of the most explored ones is the enhancement of drug efflux transporters that are responsible for reducing intracellular drug concentration [46,47,48,49]. Previous studies have associated the overexpression of subfamily B of the ATP-binding cassette members with eribulin resistance [19,20,50].

Activating the PI3K/AKT/mTOR pathway contributes to tumor development and a resistance to anticancer therapies [51]. A previous study reported that the activation of the PI3K/AKT pathway induces the primary resistance or early adaptation to eribulin in HER2-negative breast cancer models [44]. Similarly, the upregulation of the NFκB pathway increased eribulin resistance in breast cancer models [22,23]. Given that the TIMP1/CD63/PI3K/AKT/p21 axis has been described as a molecular mechanism that promotes leukemia cell proliferation and survival [52], we investigated whether *TIMP1* expression could be associated with eribulin resistance, but we detected no association (Appendix A).

In our study, the combined treatment of eribulin and elacridar significantly increased eribulin-induced apoptosis in drug-resistant leukemia cells. Interestingly, previous studies reported that elacridar improved the response to other chemotherapy agents. For example, in chronic myeloid leukemia, combining elacridar with imatinib attenuated the drug efflux transporter-associated resistance [53]. Moreover, elacridar overcame resistance to topoisomerase inhibitors in small-cell lung [54] and gastric [55] cancers. Additionally, in prostate cancer, it was demonstrated that the resistance to olaparib might also be overcome using elacridar [56]. Furthermore, the highly eribulin-resistant KBV20C oral cancer cells were shown to be sensitized by a co-treatment with a low dose of elacridar [57].

## 5. Conclusions

In summary, our data indicate that eribulin disrupts the microtubule dynamics and leads to mitotic catastrophe and cell death in blood cancer cells. The expression and activation of the MDR1, PI3K/AKT, and NFκB-related targets may be biomarkers of the eribulin response in hematologic malignancies. Additionally, the combined treatment of eribulin plus elacridar may overcome drug resistance in these diseases. Future studies must determine if eribulin can be repositioned to treat blood cancers.

## Figures and Tables

**Figure 1 cancers-14-06080-f001:**
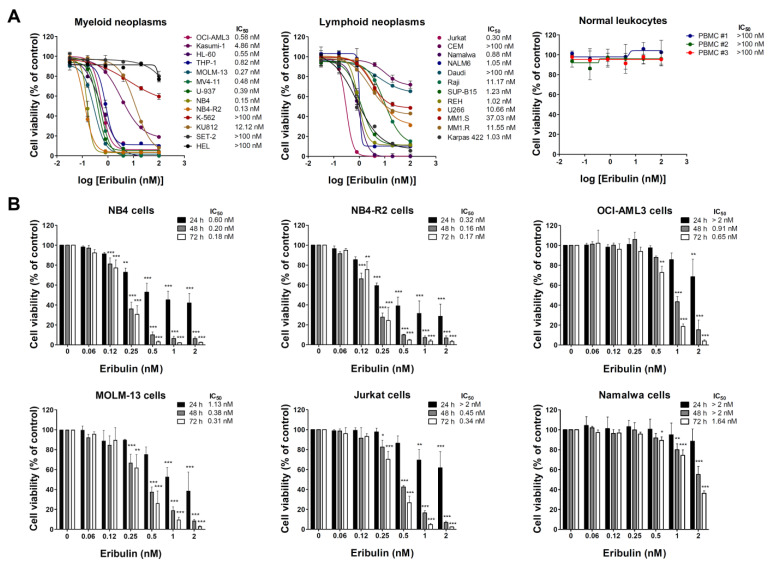
Eribulin exhibits selective dose- and time-dependent cytotoxic activity in blood cancer cells. (**A**) Dose–response curves were analyzed by methylthiazoletrazolium (MTT) assay in a panel of myeloid and lymphoid neoplasms cells or peripheral blood mononuclear cells (PBMC) from three healthy donors treated with increasing concentrations of eribulin (0.032–100 nM) for 72 h. We note that at the concentrations tested, the IC_50_ was not achieved (>100 nM) in normal leukocytes. (**B**) Bar graphs represent dose- and time-dependent responses to eribulin (0.06–2 nM) after 24, 48, and 72 h of exposure in NB4, NB4-R2, OCI-AML3, MOLM-13, Jurkat, and Namalwa cells. Values are expressed as the percentage of viable cells for each condition relative to untreated controls. Results are presented as mean ± SD of at least four independent experiments. The *p* values and cell lines are indicated in the graphs; * *p* < 0.05, ** *p* < 0.01, *** *p* < 0.001; ANOVA test and Bonferroni post-test.

**Figure 2 cancers-14-06080-f002:**
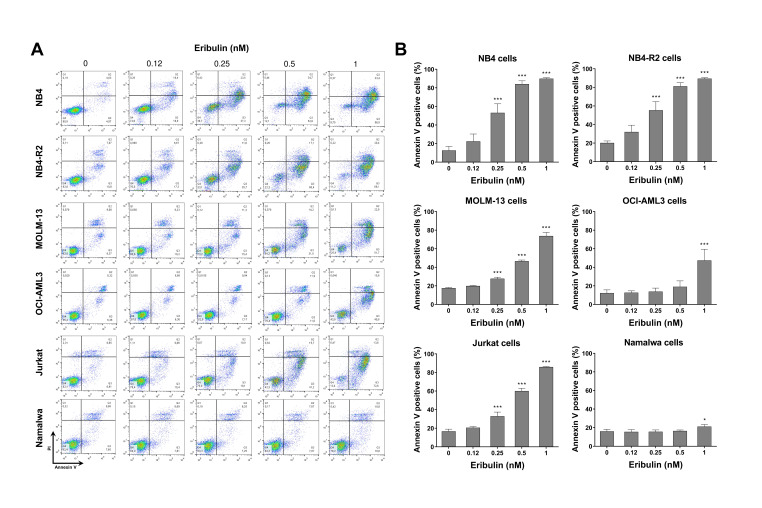
Eribulin triggers apoptosis in leukemia cells. (**A**) Apoptosis was detected by flow cytometry in NB4, NB4-R2, MOLM-13, OCI-AML3, Jurkat, and Namalwa cells treated with graded concentrations of eribulin (0.12, 0.25, 0.5, and 1 nM) for 72 h using the annexin V/propidium iodide staining method. Representative dot plots are shown for each condition; the upper and lower right quadrants (Q2 + Q3) cumulatively contain the apoptotic population (annexin V^+^ cells). (**B**) Bar graphs represent the mean ± SD of at least four independent experiments quantifying apoptotic cell death. The *p* values and cell lines are indicated in the graphs; * *p* < 0.05, *** *p* < 0.0001; ANOVA test and Bonferroni post-test.

**Figure 3 cancers-14-06080-f003:**
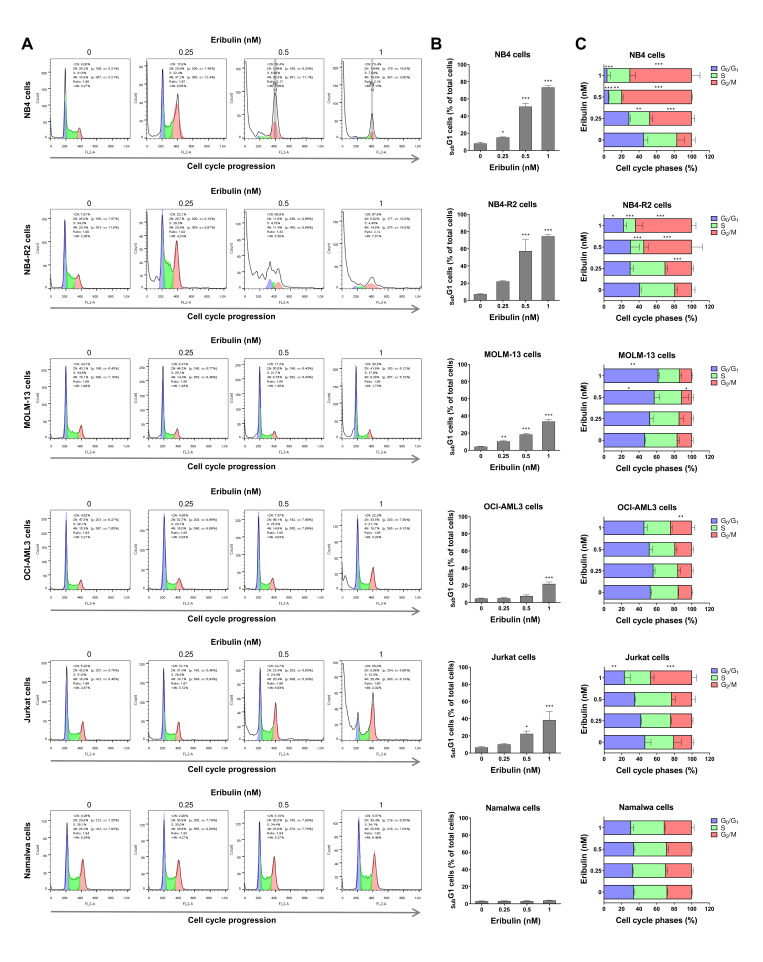
Eribulin arrests the cell cycle at G_2_/M in leukemia cells. Cell cycle phases were determined by DNA content analysis by propidium iodide staining and flow cytometry in NB4, NB4-R2, MOLM-13, OCI-AML3, Jurkat, and Namalwa cells treated with eribulin (0.25, 0.5, and 1 nM) or vehicle for 72 h. (**A**) A representative histogram for each condition is presented. (**B**) The vertical bar graph represents the mean ± SD of the cell percentages in subG_1_ from at least three independent experiments. (**C**) The horizontal bar graph represents the mean ± SD of cell distributions in the G_0_/G_1_, S, and G_2_/M phases of the cell cycle (excluding subG_1_) from at least three independent experiments. The *p* values and cell lines are indicated in the graphs; * *p* < 0.05, ** *p* < 0.01, *** *p* < 0.001; ANOVA and Bonferroni post-test.

**Figure 4 cancers-14-06080-f004:**
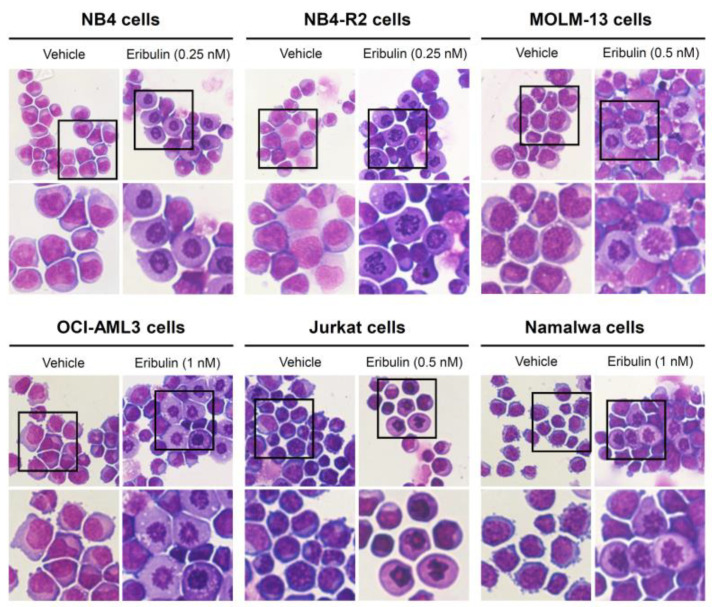
Aberrant mitoses are observed upon eribulin exposure in acute leukemia cells. NB4, NB4-R2, MOLM-13, OCI-AML3, Jurkat, and Namalwa cells were treated with vehicle or eribulin for 72 h, fixed, and stained with hematoxylin and eosin (H&E). The 400× and 1000× magnification images are displayed.

**Figure 5 cancers-14-06080-f005:**
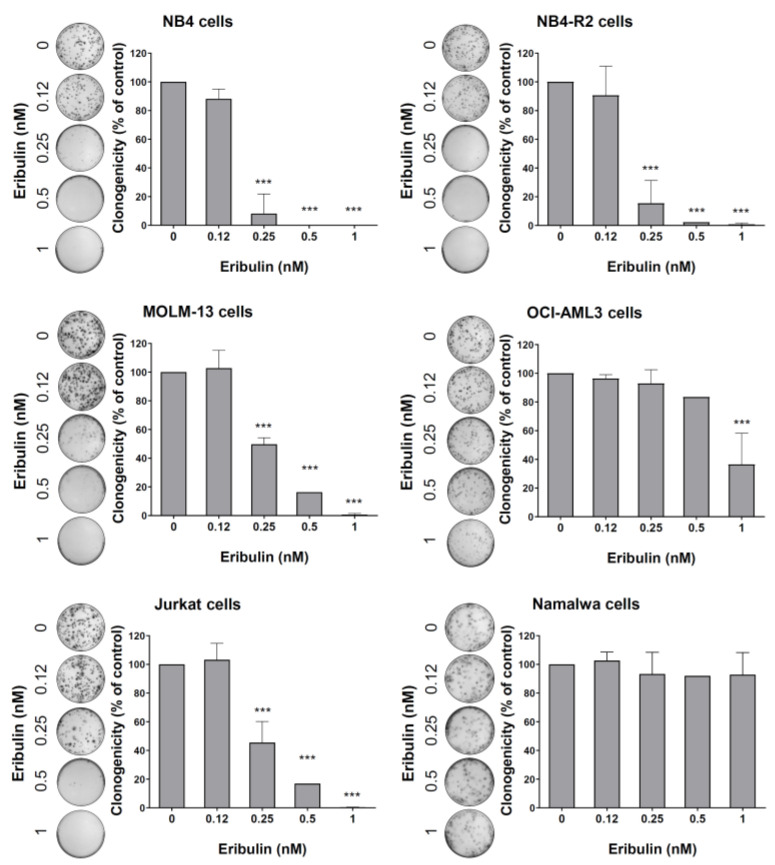
Eribulin reduces the clonogenicity of leukemia cells. Colonies containing viable cells were detected by adding an MTT reagent after eight days of culturing the cells in the presence of vehicle or eribulin (0.12, 0.25, 0.5, and 1 nM). Colony images are shown for one experiment, and bar graphs show the mean ± SD of at least three independent experiments. *** *p* < 0.0001; ANOVA test and Bonferroni post-test.

**Figure 6 cancers-14-06080-f006:**
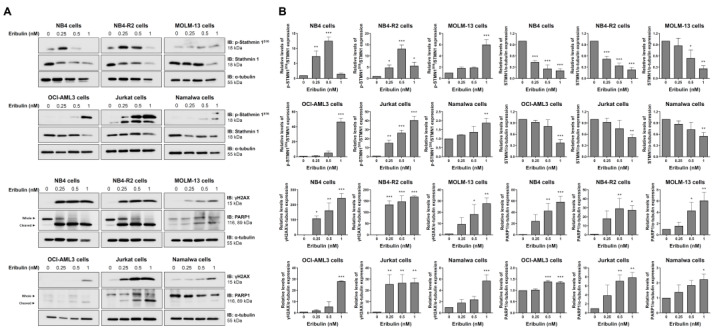
Eribulin induces molecular markers of DNA damage and apoptosis in acute leukemia cells. (**A**) Western blot analysis for levels of phospho(p)-STMN1^S16^, STMN1, γH2AX, and PARP1 (total and cleaved) in total cell extracts of vehicle- and eribulin-treated acute leukemia cells (0.25, 0.5, and 1 nM) for 48 h; Membranes were incubated with the indicated antibodies and developed with the SuperSignal West Dura Extended Duration Substrate System and images were acquired with a Gel Doc XR+. (**B**) Bar graphs represent the mean ± SD of three independent experiments quantifying the indicated protein band intensities. * *p* < 0.05, ** *p* < 0.01, *** *p* < 0.001; ANOVA and Bonferroni post-test.

**Figure 7 cancers-14-06080-f007:**
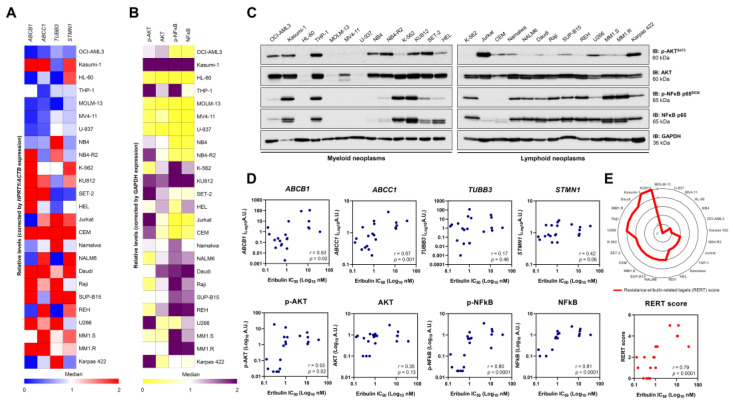
Resistance to eribulin is associated with the high expression/activation of MDR1, NFκB, and AKT in blood cancer cells. The heatmaps illustrate the expression of genes (**A**) or proteins (**B**) associated with resistance to eribulin in a panel of hematologic neoplasm cell lines. Gene data are represented as relative expression corrected by *HPRT1*/*ACTB* expression. Downregulated and upregulated genes are given by blue and red, respectively. Protein expression data are represented as relative levels corrected by the expression of GAPDH. Downregulated and upregulated proteins are indicated by yellow and purple, respectively. (**C**) Representative Western blot analysis for phospho(p)-AKT^S473^, AKT, (p)-NFκB p65^S536^, and NFκB p65 in total cell extracts from myeloid and lymphoid neoplasms cell lines. (**D**) Correlation graphs between expression of *ABCB1*, *ABCC1*, *TUBB3*, *STMN1*, p-AKT, AKT, p-NFκB, or NFκB and IC_50_ values for eribulin in blood cancer cells. (**E**) Using molecular markers that significantly correlate with IC_50_ to eribulin in hematologic neoplasms, a resistance eribulin-related targets (RERT) score was created, in which each cell line receives one point for each gene/protein upregulated (the median was used as the cutoff; the maximum number of points = 5). The radar graph shows the distribution of points among the analyzed cell lines. We note that the score precisely correlated with the drug’s sensitivity in the cellular models evaluated.

**Figure 8 cancers-14-06080-f008:**
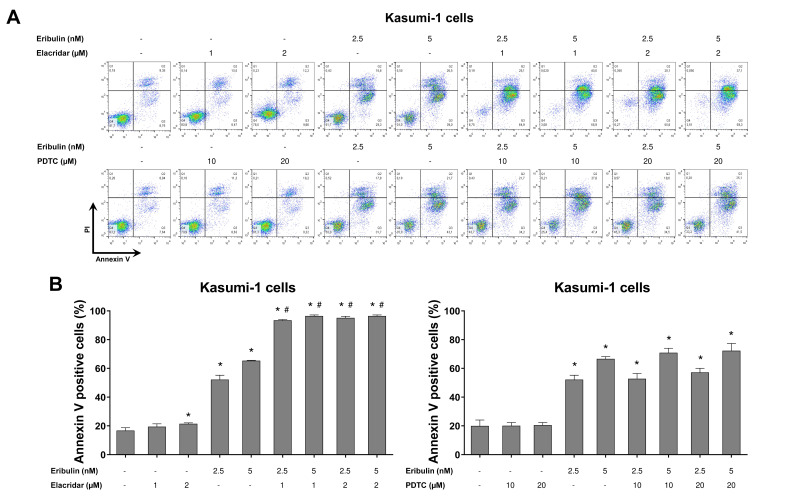
Elacridar, a P-glycoprotein inhibitor, potentiates eribulin-induced apoptosis in Kasumi-1 cells. (**A**) Apoptosis was detected by flow cytometry using APC-annexin V and propidium iodide staining. Representative dot plots are counters for each condition; the upper and lower right quadrants cumulatively the apoptotic population (annexin V^+^ cells). (**B**) Bar graphs represent the mean ± SD of at least three independent experiments quantifying apoptotic cell death in Kasumi-1 cells after exposure to the vehicle, eribulin (2.5 and 5 nM) and/or elacridar (1 and 2 µM) and/or PDTC (10 and 20 µM) for 72 h. The *p* values are indicated in the graphs; * *p* < 0.05 for eribulin-, elacridar-, and/or PDTC-treated cells vs. vehicle-treated cells, # *p* < 0.05 for eribulin-, elacridar-, or PDTC-treated cells versus combination treatment at the corresponding doses; ANOVA and Bonferroni post-test.

## Data Availability

The datasets used and/or analyzed during the current study are available from the corresponding author upon reasonable request.

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
