# Peer review of "Cellular and Molecular Effects of Eribulin in Preclinical Models of Hematologic Neoplasms"

_cancers, 2022, doi:10.3390/cancers14246080_

Round 1
Reviewer 1 Report
In the article “Cellular and Molecular Effects of Eribulin in Preclinical Models of Hematologic Neoplasms” Vicari et al., provided evidence of the effects of eribulin on hematologic cancers, as well as the identification of potential biomarkers of responsiveness.
The work is interesting and has translational value. I have the following recommendations to improve the clarity.
Comments:
1. Did authors see RNA-seq analysis on different AML cell lines treated with eribulin to see differential gene expression related to microtubule?
2. Immunofluorescence studies were more convincing to show mitotic aberrations upon treatment with eribulin, rather than H&E staining.
3. It is known that TIMP-1 modulates leukemic cell survival, migration and function via CD63/PI3K/Akt/p21 signaling and intracellular accumulation of CD63, a tetraspanin component of small extracellular vesicles, in late/multivesicular endosomes of triple-negative breast cancer cells. Is it possible that TIMP-1 or CD63 accumulation is involved in eribulin resistance in hematologic malignancies?
Author Response
Reviewer #1
1- Did authors see RNA-seq analysis on different AML cell lines treated with eribulin to see differential gene expression related to microtubule?
Authors' response: We would like to thank the Reviewer for the suggestion and the opportunity to clarify this point. Unfortunately, we do not have RNAseq data from leukemia cells treated with eribulin. As our study is pioneering, no data has been deposited in this context. Performing RNAseq without prior planning is not trivial in our country due to the high costs involved. However, this is an interesting perspective for the future continuity of this project.
2- Immunofluorescence studies were more convincing to show mitotic aberrations upon treatment with eribulin, rather than H&E staining.
Authors' response: In previous works, we performed immunofluorescence at the request of Reviewers to assess issues associated with microtubule dynamics and mitosis in leukemia cells, and the results were always uninformative. Leukemia cells have little cytoplasm, and adhesion to the coverslip is done with poly-L-lysine, which generally retains a high staining background and poor resolution. Due to poor adhesion, the number of cells retained at the end of staining for analysis is limited. Therefore, we would like the reviewer to consider that the H&E assays only illustrate the observed morphology but that the quantification of cell cycle aberrations was performed by flow cytometry.
3- It is known that TIMP-1 modulates leukemic cell survival, migration and function via CD63/PI3K/Akt/p21 signaling and intracellular accumulation of CD63, a tetraspanin component of small extracellular vesicles, in late/multivesicular endosomes of triple-negative breast cancer cells. Is it possible that TIMP-1 or CD63 accumulation is involved in eribulin resistance in hematologic malignancies?
Authors' response: The choice of eribulin-associated biomarkers was based on previous studies in solid tumors, which presented clinical or functional validation of genes/proteins associated with drug response. However, the reviewer's comment intrigued us and encouraged us to test the hypothesis raised. We investigated TIMP1 expression in this context. However, we found no significant correlations between TIMP1 expression and response to eribulin. We have added this information to the discussion and supplemental material of the revised version of the manuscript.

Reviewer 2 Report
The authors report cellular and molecular effects of eribulin in preclinical models of hematologic neoplasms.
1. In Figure 7, pAkt and pNFκB were associated with IC50 values for eribulin. However, HEL cell line is highly resistant to eribulin (>100nM), pAkt and pNFκB were not increased by immunoblot analysis (Figure 7c). THP-1 is sensitive to eribulin but pAkt and pNFκB were increased like Kasumi-1. Thus, pAkt and pNFκB may not be associated with IC50 values for eribulin.
2. The authors should provide the data of eribrin plus elacridar against normal leukocytes.
Author Response
Reviewer #2
1- In Figure 7, pAkt and pNFκB were associated with IC50 values for eribulin. However, HEL cell line is highly resistant to eribulin (>100nM), pAkt and pNFκB were not increased by immunoblot analysis (Figure 7c). THP-1 is sensitive to eribulin but pAkt and pNFκB were increased like Kasumi-1. Thus, pAkt and pNFκB may not be associated with IC50 values for eribulin.
Authors' response: The authors thank the reviewer for the opportunity to discuss and clarify this issue. As expected, as with most biomarkers of response to therapy, none are 100% effective in predicting response, and there are always exceptions to the rule. Thus, our conclusions are based on the global view involving 25 cell lines and statistical analysis. The examples cited by the Reviewer are exceptions but do not invalidate the hypothesis. Due to this work's pioneering nature in using eribulin in hematological neoplasms, these biomarkers were chosen based on the knowledge generated in solid tumors. With these data, we can better delineate future investigations into biomarkers for eribulin response in blood cancers.
2- The authors should provide the data of eribrin plus elacridar against normal leukocytes.
Authors' response: The authors thank the Reviewer for the excellent suggestion. We analyzed the effects of eribulin plus elacridar against normal leukocytes from five healthy independent donors. The treatment in monotherapy or combination did not show exacerbated toxicity in any of the conditions evaluated (the same used for the leukemia cell line, Kasumi-1).

Round 2
Reviewer 2 Report
none